# Microwave Heating of the Catalyst Bed as a Way of Energy-Saving Oxidative Dehydrogenation of Ethane on a Mo-V-Te-Nb-O_x_ Catalyst

**DOI:** 10.3390/nano12244459

**Published:** 2022-12-15

**Authors:** Alexei V. Kucherov, Nikolai A Davshan, Elena D. Finashina, Leonid Kustov

**Affiliations:** 1N.D. Zelinsky Institute of Organic Chemistry, RAS, Leninsky pr. 47, 119991 Moscow, Russia; 2Department of Chemistry, Lomonosov Moscow State University, 119992 Moscow, Russia; 3National University of Science and Technology MISiS, Leninsky Prosp. 4, 119991 Moscow, Russia

**Keywords:** catalyst Mo-V-Te-Nb-Ox, microwave heating, oxidative dehydrogenation, ethane, ethylene

## Abstract

In search of a more effective process of ethane oxidative hydrogenation, different operation modes (thermal and microwave heating) are compared. The catalyst Mo_1_-V_0.3_-Te_0.13_-Nb_0.11_-O_x_ was prepared by hydrothermal synthesis and characterized by a set of physicochemical methods (XRD, N_2_ adsorption, SEM, EDX). The direct microwave heating of the catalyst layer is proposed as an alternative way of energy-saving ethane-to-ethylene oxidation by a Mo-V-Te-Nb-O_x_ system. A substantial decrease in the reactor temperature upon the microwave-assisted process is accompanied by extremely high catalyst selectivity, which remains at a very high level of 98+%.

## 1. Introduction

Catalytic dehydrogenation processes are widely used in modern oil refining and petrochemistry. Dehydrogenation catalysts occupy third place in the industrial production of catalysts in terms of production and use after cracking and hydrotreating catalysts. The dehydrogenation of lower paraffins (C_2_–C_4_) is a significant problem, which is associated with the low reactivity of alkanes, the reversibility of the non-oxidative dehydrogenation process, high process temperatures, low selectivity, and stability of catalysts in the case of oxidative dehydrogenation. The use of oxidative dehydrogenation allows one to significantly reduce the temperature of the process. Due to the growing demand for the processing of natural and associated gases of oil refining, the issue of developing new catalysts and oxidative dehydrogenation technologies that will allow converting the main components of these gases—ethane, propane, and butane—into corresponding olefins is acute [1].

Several alternatives attract the attention of researchers in search of the improvement of thermal catalytic processes, including microwave and plasma technologies and the application of an external electric field [2,3,4,5,6]. Earlier, we tested the latter approach in the study of the electro-heating of a Mo-V-Te(Fe)-Nb-O_x_ system that is active in the oxidative dehydrogenation of ethane (ODE) [7]. It was concluded that the peculiarities of ODE in this system upon the electrical treatment of the catalyst layer can be explained by thermal heating without additional electronic effects. 

The advantages of processing various catalysts in the microwave field in comparison with thermal activation have been demonstrated previously for several processes. In some cases, a considerable decrease in the onset temperature of many catalytic processes due to the formation of hot spots in the bulk volume of the catalyst was observed, as well as non-trivial changes in the selectivity of the catalytic processes associated with non-equilibrium microwave heating conditions being caused by the microwave field and non-thermal effects [8,9,10,11,12,13,14,15,16].

A quaternary oxide system, Mo-V-Te-Nb-O_x_, was widely investigated due to its unique ability to selectively catalyze ODE [17,18,19,20,21,22,23,24]. From the electro-conductivity viewpoint, these mixed oxides demonstrate semiconductor properties, with a rather high conductivity at temperatures of the catalytic process. On the other side, this system demonstrates a strong interaction with microwave (*mw*) radiation. The effective absorption of the *mw*-power was confirmed by our preliminary testing of the catalyst Mo-V-Te-Nb-O_x_. Therefore, this sample seemed to be suitable for studying the catalytic process upon direct *mw*-treatment. For the reaction under study, Te-containing four-component oxide looks like the best selective catalytic system. Attempts to change the chemical composition or add some doping agents to the catalyst’s composition have not been successful until now [25,26]. 

A catalytic bed consisting of small particles of the pure catalyst provides good permeability for cold gas flow, and microwave radiation supports heat release directly inside the catalytic particles. Therefore, the idea of this work was to monitor a possible role of the in situ microwave treatment of the Mo-V-Te-Nb-O_x_ catalyst bed in ODE in search of a possible improvement of the process activity/selectivity. 

## 2. Results

### 2.1. Catalyst Characterization

The crystallinity and phase composition of the catalyst was determined by X-ray diffraction (2θ range of 5° to 60°). XRD analysis of the catalyst was obtained (Figure 1), and post-catalytic tests are identical and indicate the presence of a well-crystallized mixture of M1 and M2 phases in accordance with the literature data [27]. It is known from the literature that the active and selective catalysts of the MoVTeNb composition are a mixture of two crystalline phases (M1 + M2) and may also contain a small number of amorphous impurities [17].

The N_2_ adsorption isotherms measured at −196 °C for the catalysts are presented in the Appendix A. The surface area of the catalyst determined by the BET method (Brunauer–Emmett–Teller) is 9.1 m^2^/g, which is also consistent with the literature data [17,27]. The sample contains mainly mesopores. The mesopore size distribution for the Mo-V-Te-Nb-O catalyst (calculation by the BJH (Barrett, Joyner, and Halenda) method, desorption branch of the isotherm) is shown in Appendix A. The shape of the N_2_ adsorption isotherm at high relative pressures shows that there are macropores in the sample that are not determined by nitrogen adsorption.

The elemental composition of the catalysts obtained by the EDX method is presented in Table 1. A micrograph of the sample is presented in Figure 2.

The study of the catalyst surface by the EDX method shows that its composition is fairly homogeneous. In the three selected areas, the maximum standard deviation of the composition for four elements (Mo, V, Te, and Nb) is 0.64 at. %.

According to the results of the EDX study, the composition of the catalyst corresponds to the composition determined by the ratio of precursors at the preparation stage: MoV_0.32_Nb_0.11_Te_0.13_O_4.25_. EDX mapping has revealed a uniform distribution of elements over the crystal phase of catalysts (Appendix A).

The XPS analysis demonstrates the presence of Mo, Nb, V, Te, and O. The calculated formula of the catalyst composition (per 1 Mo atom) is MoV_0.30_Nb_0.11_Te_0.13_O_4.29_. XPS spectra in the regions characteristic of Mo, V, and Nb are presented in Appendix A. Table 2 summarizes the binding energies and oxidation states of the elements in the MoVTeNbO catalyst. The results obtained show that all elements are in the highest oxidation states. After treatment with microwave radiation, the oxidation state of the elements in the catalyst does not change (Appendix A).

It should be noted that there is a good agreement between the results of determining the composition of the catalyst obtained using “surface science” methods (EDX, XPS), as well as the composition of the catalyst, determined using the ICP OES method.

### 2.2. Catalyst Testing

The reaction under study is exothermic, and heat release inside the bed of the undiluted catalyst working in the undiluted gas mixture can cause some additional rise in the temperature. However, the use of the embedded thermocouple permits us to measure quite precisely the average temperature inside the catalyst bed, as it was demonstrated in the preliminary testing of the catalyst *mw*-heating in air. Therefore, the results of the catalytic testing of the Mo-V-Te-Nb-O_x_ sample with *mw*-heating in the quartz reactor can be presented as temperature dependences (Figure 3 and Figure 4, Table 3). The results of conventional testing, with the heating of the sample by an outer oven, are presented for comparison in Figure 5. The long-term testing of the catalyst demonstrated that after 10 h of work, the conversion of ethane decreases from 30 to 28% without any selectivity loss.

## 3. Discussion

The results of the physico-chemical study indicate that *mw*-irradiation does not affect the catalyst structure (phase composition, crystallinity, surface area, element composition, and oxidation state). As to the thermal treatment, some destruction of the catalyst was observed at temperatures above 450 °C. 

A comparison of the conversion curves (Figure 3a, Figure 4a and Figure 5a) demonstrates a rather strong positive effect of *mw*-heating: the catalytic process is shifted by ~100 °C to lower temperatures. Therefore, the process of the oxidative dehydrogenation of ethane to ethylene on the Mo-V-Te-Nb-O_x_ catalyst can be realized under milder conditions. 

However, it is necessary to take into account that the *mw*-heating of the granulated layer of *mw*-absorbing material, Mo_1_V_0.3_Te_0.13_Nb_0.11_O_x_, causes heat recovery directly inside the catalyst particles, with further heat dissipation to the cold gas flowing inside the reactor. In other words, the system under investigation demonstrates the mosaic distribution of temperatures throughout the bed, and the embedded thermocouple measures some averaged (integrated) values (Figure 3 and Figure 4). As a result, a direct comparison of these data with results obtained under the conditions of a uniform temperature field (Figure 5) is complicated. 

On the other hand, the properties of the samples working in different heating conditions can be compared qualitatively by plotting the results as a dependence of the activity/selectivity vs. ethane conversion (Table 2). From these data, one can see that in cases of *mw*-heating, the rise in the conversion is accompanied by a gradual loss in the selectivity being quite similar to one observed upon external oven heating (Table 2). Consequently, even when *mw*-heating the pure Mo_1_V_0.3_Te_0.12_Nb_0.15_O_x_, no measurable improvement of the process selectivity can be achieved

## 4. Materials and Methods

### 4.1. Catalyst Preparation

The 4-component Mo_1_V_0.3_Te_0.13_Nb_0.11_O_x_ sample was synthesized hydrothermally according to Finashina et al. [18]. Washed and dried powder was calcined in pure N_2_ at 600 °C, then pressed, crushed, and sieved to obtain the fraction 0.3–0.5 mm. The charge of the sample (~0.3 cc, 370 mg) was placed for catalytic testing in the middle of a quartz tube (inner diameter 7 mm) and plugged from both sides by quartz wool. After the first set of catalytic testing, the catalyst charge was unloaded, mixed with a portion of crushed quartz (same fraction 0.3–0.5 mm; 1:1 by volume), and placed back into the micro-reactor for repeated catalytic testing. 

### 4.2. Catalyst Testing

The aim of this work was to perform a comparative study in the ODE of pure and diluted Mo-V-Te-Nb-O_x_ catalysts under conditions of conventional thermal or microwave heating. 

To conduct the reaction under conditions of microwave heating, the quartz reactor was placed into a cylindrical metallic resonator (working frequency ~3.8 GHz) connected to a laboratory microwave setup (maximum output power 50 W). The use of the resonator device permitted us to concentrate *mw*-power inside the reactor zone, providing extremely high efficiency of power absorption (close to 98%). Therefore, the catalyst bed was heated by *mw*-power in the middle of the resonator connected by waveguides with a G4-80 microwave generator (2.5–4.0 GHz range of working frequencies) and power amplifier. The average temperature inside the catalyst layer was measured using a thin embedded thermocouple connected to an M5-78V thermoelectric converter. The *mw*-heating of the thermocouple itself in the empty reactor was negligible. The temperature of the catalyst was regulated by varying both the microwave radiation frequency and power.

As to the gas flow, the catalyst was tested at an atmospheric pressure in the gas mixture [75% C_2_H_6_ + 25% O_2_] at the gas flow rate of 2200–2500 h^−1^. For this gas mixture, the conversion of ethane theoretically can reach ~66% (at 100% selectivity and complete O_2_ consumption). In our tests, the ethane conversion was kept at a level <40% to avoid the complete removal of the oxidant from the gas phase. 

The probes of the outgoing flow were analyzed by gas chromatography. Ethane, ethylene, water, O_2_, and CO_2_ were detected as the main components of the mixture, with a trace admixture of acetic acid. The undiluted catalyst (mw-1) was tested first, and then measurements were repeated with the sample diluted by quartz (1:1 by vol.) (mw-2). 

### 4.3. Catalyst Characterization

XRD. The phase composition of the materials was studied by X-ray diffraction (XRD) analysis. X-ray diffraction patterns were recorded using an ARL X’TRA diffractometer (Thermo Fisher Scientific, Basel, Switzerland) with CuKα radiation (40 kV, 40 mA) with a scanning rate of 1.2° per minute over the scanning range of 5 < 2θ < 60°. ICCD data were used for the identification purpose. 

N_2_ adsorption−desorption analysis was conducted using an ASAP 2020 Accelerated Surface Area and Porosimetry instrument (Micromeritics, Unterschleißheim, Germany) equipped with an automated surface area measurement unit at −196 °C using BET calculations for the surface area.

SEM-EDX. The morphology, particle size, and elemental composition on the catalyst surface were studied via scanning electron microscopy (SEM) using an LEO EVO 50 XVP electron microscope (Carl Zeiss, Aalen, Germany) equipped with an INCA Energy 350 energy dispersive spectrometer (Oxford Instruments, Abingdon, Great Britain).

*XPS*. The oxidation states of the elements in the MoVTeNbO catalyst were determined by XPS. The spectra were measured using a Kratos Axis Ultra DLD spectrometer with Al-Kα irradiation (E = 1486.6 eV). The C1s (285.0 eV) line was used for calibration.

ICP-OES. The contents of the elements in the catalysts were determined by inductively coupled plasma-optical emission spectrometry (ICP-OES) with a Varian 715-ES spectrometer after the dissolution of the solid sample in an acid mixture of HNO_3_:HF:HCl in a 1:1:3 volume ratio.

## 5. Conclusions

Thus, an energy-saving approach with the *mw*-heating of the catalytic bed provides a considerable lowering of the averaged reactor temperature but is unable to change the selectivity of the catalyst working in a non-uniform temperature field.

## Figures and Tables

**Figure 1 nanomaterials-12-04459-f001:**
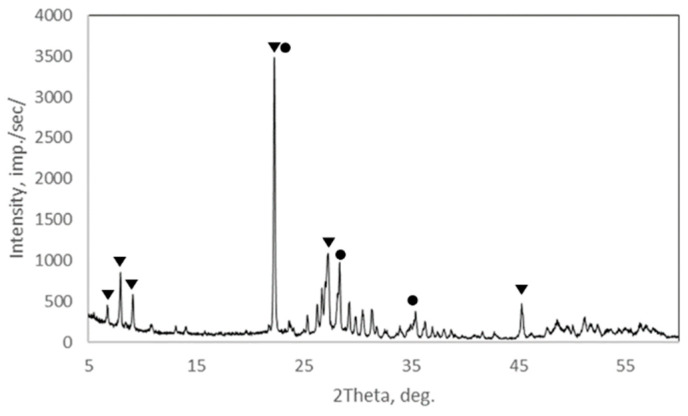
X-ray diffractogram of the Mo-V-Te-Nb catalyst. 
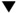
—characteristic reflexes of the orthorhombic phase M1; 
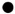
—characteristic peaks of the hexagonal phase (M2).

**Figure 2 nanomaterials-12-04459-f002:**
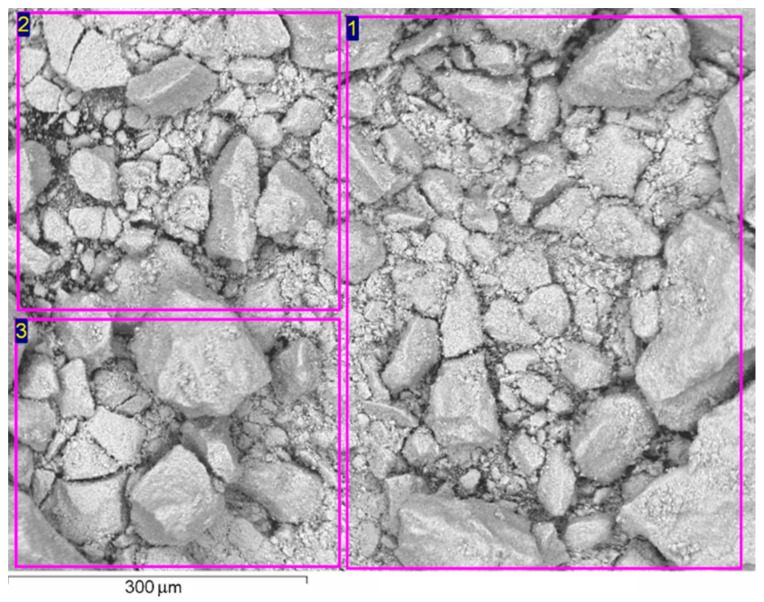
A micrograph of the catalyst sample. The selected areas (1–3) on the surface of the catalyst were used to determine the composition by the EDX method and correspond to spectra 1–3 in Table 1.

**Figure 3 nanomaterials-12-04459-f003:**
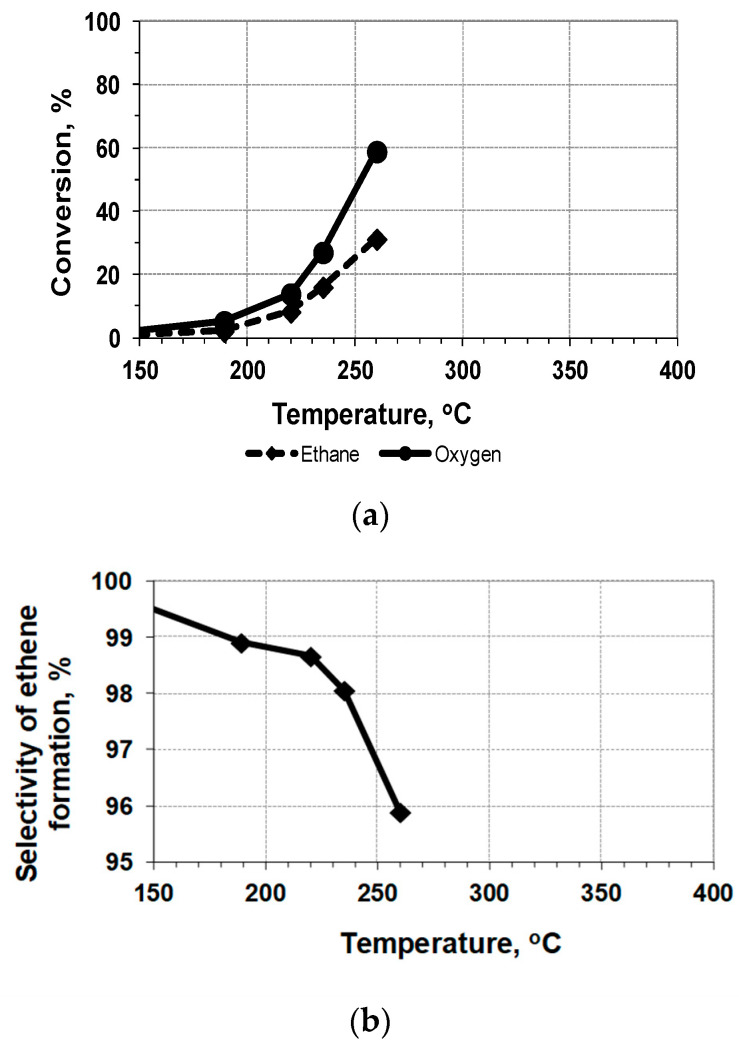
ODE with *mw*-heating of the undiluted sample Mo-V-Te-Nb-O_x_ (0.37 g; 0.3 cc): (**a**)—conversion of reagents; (**b**)—selectivity of C_2_H_4_ formation; gas mixture [75%C_2_H_6_ + 25%O_2_], flow rate = 660 cm^3^/h, GHSV = 2200 h^−1^.

**Figure 4 nanomaterials-12-04459-f004:**
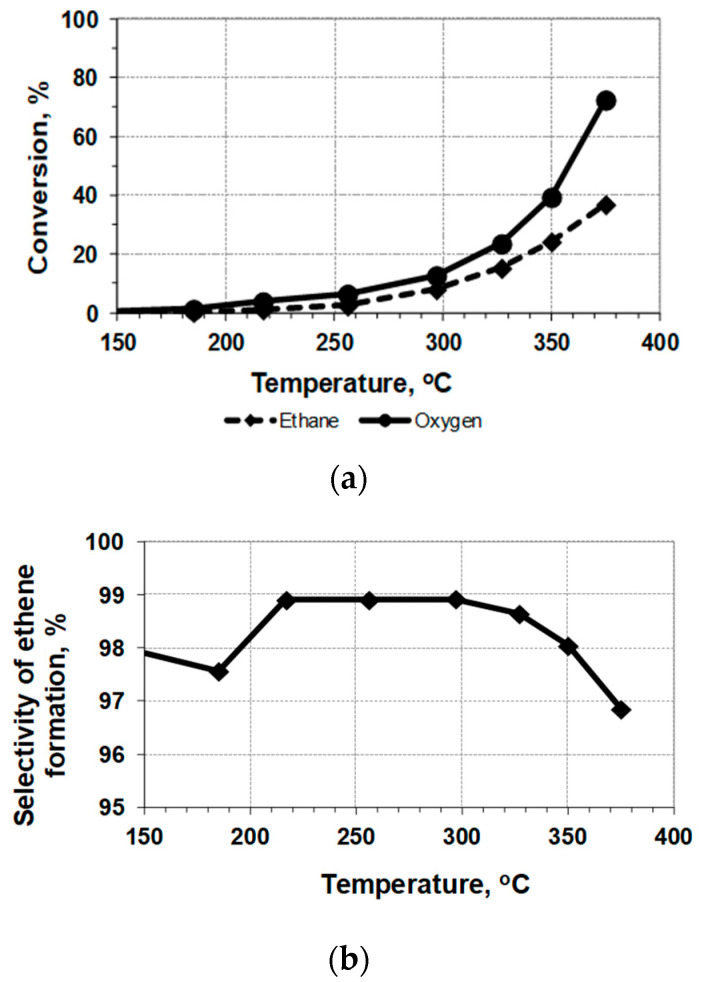
ODE with *mw*-heating of the same sample Mo-V-Te-Nb-Ox (0.37 g; 0.3 cc) diluted by quartz (1/1): (**a**)—conversion of reagents; (**b**)—selectivity of C_2_H_4_ formation; gas mixture [75%C_2_H_6_ + 25%O_2_], flow rate = 660 cm^3^/h, GHSV = 2200 h^−1^.

**Figure 5 nanomaterials-12-04459-f005:**
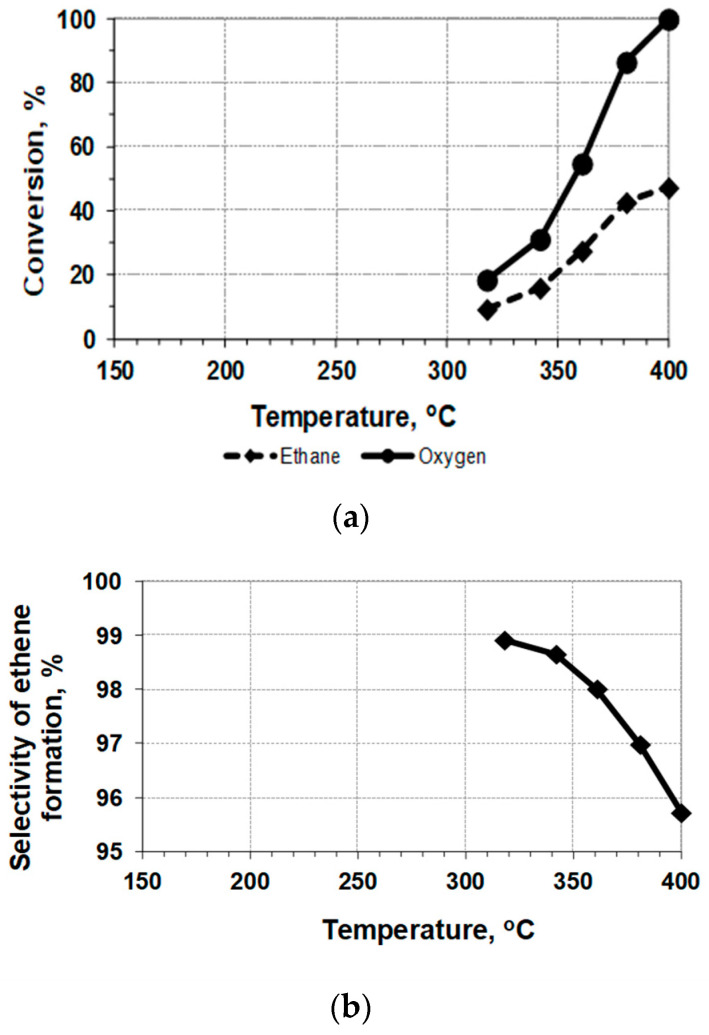
ODE with oven-heating of the sample Mo-V-Te-Nb-O_x_ (0.144 g; 0.12 cc) diluted by quartz (1/1): (**a**)—conversion of reagents; (**b**)—selectivity of C_2_H_4_ formation; gas mixture [78%C_2_H_6_ + 22%O_2_], flow rate = 600 cm^3^/h, GHSV = 2500 h^−1^.

**Table 1 nanomaterials-12-04459-t001:** The elemental composition of the catalysts obtained by the EDX method (at. %).

Spectrum	Mo	V	Te	Nb	O
1	17.88	5.72	2.25	1.82	72.32
2	17.15	5.59	2.29	1.83	73.13
3	16.61	5.33	2.24	1.72	74.10
Average	17.21	5.55	2.26	1.79	73.18
Standard deviation	0.64	0.20	0.03	0.06	0.89
Max.	17.88	5.72	2.29	1.83	74.10
Min.	16.61	5.33	2.2	1.72	72.32

**Table 2 nanomaterials-12-04459-t002:** Binding energies and oxidation states of the elements in the MoVTeNbO catalyst.

XPS Line	Binding Energy, eV	State
Mo 3d_5/2_	233.2	Mo^+6^
Nb 3d_5/2_	207.4	Nb^+5^
V 2p_3/2_	517.1	V^+5^
Te 3d_5/2_	577.0	Te^+6^

**Table 3 nanomaterials-12-04459-t003:** Temperatures of X%-conversion of ethane (T) and corresponding values of selectivity of ethylene formation (S) on the catalyst Mo_1_V_0.3_Te_0.13_Nb_0.11_O_x_ tested in different conditions.

X, %	Oven Heated		mw-1 (Pure)		mw-2 (Diluted)	
	T, ^0^C	S, %	T, ^0^C	S, %	T, ^0^C	S, %
10	320	98.8	223	98.7	305	98.8
15	338	98.6	233	98.2	328	98.6
20	348	98.4	242	97.5	340	98.2
25	357	98.1	250	96.8	351	97.9
30	364	97.8	258	96.0	362	97.7
35	371	97.3	267	95.2	370	97.0

## Data Availability

Data are available upon request.

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
