# Peer review of "Microwave Heating of the Catalyst Bed as a Way of Energy-Saving Oxidative Dehydrogenation of Ethane on a Mo-V-Te-Nb-Ox Catalyst"

_nanomaterials, 2022, doi:10.3390/nano12244459_

Round 1

Reviewer 1 Report

Nanomaterials

Microwave heating of the catalyst bed as a way of energy-saving oxidative dehydrogenation of ethane on a Mo-V-Te-Nb-Ox catalyst

Comment to the Authors

Products of oxidative dehydrogenation of ethane have tremendous application in chemical synthesis and petrochemical industry and therefore, it is a significant topic of interests globally. Finding cost effective, low energy alternative options by fine tuning catalyst for the process is widely studied and such means would also be environment friendly. In this interesting article authors Alexei Kucherov, Nikolai Davshan, Elena Finashina and Leonid Kusov studied thermal and microwave modes to for effective ethane oxidative hydrogenation. Mo-V-Te-Nb-Ox catalyst was used with direct microwave heating of catalyst layer to conserve energy in ethane to ethylene oxidation process with high selectivity. It is very remarkable to see significant reduction of temperature after microwave heating of the undiluted catalyst. Authors have shown their findings. However, for each observation and results authors didn’t provide their comments or discussion. Such discussions would be very useful for the researchers in the field of catalysis, chemical synthesis and for the broad readers of Nanomaterials journal. Introductions, results, discussions, and conclusion sections need to be modified to increase significance of this study.

Findings provided are reasonable but need further discussion to make it more informative for the readers. The following points needs to be addressed before publication, to make this article more helpful for the readers of Nanomaterials:

1.     It would be informative for readers if authors include few comments about ethane oxidative hydrogenation process and its significance in the introduction. What are the difficulties and disadvantage of this process along with thermal disadvantage. What are the current available catalysts for this process other than Mo-V-Te(Fe)-Nb-Ox, and what are their advantages and disadvantages. What are the advantages and disadvantages or other catalytic processes? Are there any biocatalysts available for this process? This will further help to increase the significance of the topic and would be interesting for broad range readers.

2.     Figure 2 on page 2 is not well labeled. What are three windows 1,2 and 3 illustrates? Needs to be labeled in figure or in caption.

3.     It would be helpful if authors add comments for their findings and observations for the results of EDX study shown in Figure 2 and Table 1. Later in the text authors have added discussion section. However, adding few comments followed after each observation and results illustrated in each Figure and Table would be easier for the readers to understand the results better.

4.     Please provide reference for the claim, “However, the use of embedded thermocouple permits one to measure quite precisely the average temperature inside the catalyst bed” page 3 line 81-82.  

5.     Have authors also studied the time factor (turnover values) for each of these process for the conversion? In Table 2 authors considered only % conversions up to 30%. If authors add more data points up to 50% (as shown in Figures 3, 4 and 5) of conversion of ethane would be very helpful to see the temperature and selectivity for each process.

6.     Authors mentioned that microwave heating the Mo-V-Te(Fe)-Nb-Ox catalyst is efficient way for energy saving. However, the catalyst containing elements are costly and not environmentally friendly. Authors comments on how to address such issues in future studies in conclusion would be valuable.

Minor corrections:  

i                  Please define all jargon for broad readers, for e.g., Page 2 Line 63: BET and line 65: BJH

ii                 Page 2 line 66: typo “adsorbtion”  

iii               SI file Figure S1: Legend is missing in the figure for the two different graphs.

Author Response

Comment: Products of oxidative dehydrogenation of ethane have tremendous application in chemical synthesis and petrochemical industry and therefore, it is a significant topic of interests globally. Finding cost effective, low energy alternative options by fine tuning catalyst for the process is widely studied and such means would also be environment friendly. In this interesting article authors Alexei Kucherov, Nikolai Davshan, Elena Finashina and Leonid Kusov studied thermal and microwave modes to for effective ethane oxidative hydrogenation. Mo-V-Te-Nb-Ox catalyst was used with direct microwave heating of catalyst layer to conserve energy in ethane to ethylene oxidation process with high selectivity. It is very remarkable to see significant reduction of temperature after microwave heating of the undiluted catalyst. Authors have shown their findings. However, for each observation and results authors didn’t provide their comments or discussion. Such discussions would be very useful for the researchers in the field of catalysis, chemical synthesis and for the broad readers of Nanomaterials journal. Introductions, results, discussions, and conclusion sections need to be modified to increase significance of this study.

Response: We would like to thank the reviewer for the critical comments and useful advices and questions raised. We tried to do our best to improve our manuscript by taking into account all the comments. Below we give our responses to the comments. The changes in the text are highlighted in yellow. More data and relevant discussions are added in the corresponding parts of the article. 

Comment: Findings provided are reasonable but need further discussion to make it more informative for the readers. The following points needs to be addressed before publication, to make this article more helpful for the readers of Nanomaterials:

Response: Done.

Comment: It would be informative for readers if authors include few comments about ethane oxidative dehydrogenation process and its significance in the introduction. What are the difficulties and disadvantage of this process along with thermal disadvantage. What are the current available catalysts for this process other than Mo-V-Te(Fe)-Nb-Ox, and what are their advantages and disadvantages. What are the advantages and disadvantages or other catalytic processes? Are there any biocatalysts available for this process? This will further help to increase the significance of the topic and would be interesting for broad range readers.

Response: Some comments were added to the Introduction part of the article. No biocatalysts for the oxidative dehydrogenation of ethane are known in the art. Catalytic dehydrogenation processes are widely used in modern oil refining and petrochemistry. Dehydrogenation catalysts occupy the third place in the industrial production of catalysts in terms of production and use after cracking and hydrotreating catalysts. The dehydrogenation of lower paraffins (C2-C4) is a significant problem, which is associated with the low reactivity of alkanes, the reversibility of the non-oxidative dehydrogenation process, high process temperatures, low selectivity and stability of catalysts in the case of oxidative dehydrogenation. The use of oxidative dehydrogenation allows one to significantly reduce the temperature of the process.  Due to the growing demand for processing of natural and associated gases of oil refining, the issue of developing new catalysts and oxidative dehydrogenation technologies that will allow converting the main components of these gases – ethane, propane and butane into corresponding olefins is acute.

Comment: Figure 2 on page 2 is not well labeled. What are three windows 1,2 and 3 illustrates? Needs to be labeled in figure or in caption.

Response: The micrograph shows the areas of the catalyst surface in which the probe was placed to determine the composition by the EDX method. The necessary comments have been added to the figure 3 caption.

Comment: It would be helpful if authors add comments for their findings and observations for the results of EDX study shown in Figure 2 and Table 1. Later in the text authors have added discussion section. However, adding few comments followed after each observation and results illustrated in each Figure and Table would be easier for the readers to understand the results better.

Response:  Some comments are added (page 3). The study of the catalyst surface by the EDX method shows that its composition is fairly homogeneous. In the three selected areas, the maximum standard deviation of the composition for four elements (MoVTeNb) is 0.64 at%.

Comment: Please provide reference for the claim, “However, the use of embedded thermocouple permits one to measure quite precisely the average temperature inside the catalyst bed” page 3 line 81-82. 

Response: The phrase is modified: “However, the use of the embedded thermocouple permits us to measure quite precisely the average temperature inside the catalyst bed, as it was demonstrated in our preliminary testing of the catalyst heating by mw-power on air”.

Comment: Have authors also studied the time factor (turnover values) for each of these process for the conversion? In Table 2 authors considered only % conversions up to 30%. If authors add more data points up to 50% (as shown in Figures 3, 4 and 5) of conversion of ethane would be very helpful to see the temperature and selectivity for each process.

Response:  One more set of data is added to Table 2 (conversion 35%). For the gas mixture used, the conversion of ethane theoretically can reach ~66% (at 100% selectivity and complete O2 consumption). In our tests, the ethane conversion was kept on the level <40% to avoid the complete removal of the oxidant from the gas phase. This explanation is added to the text.

Comment: Authors mentioned that microwave heating the Mo-V-Te(Fe)-Nb-Ox catalyst is efficient way for energy saving. However, the catalyst containing elements are costly and not environmentally friendly. Authors comments on how to address such issues in future studies in conclusion would be valuable.

Response: For the reaction under study, Te-containing four-component oxide looks the best selective catalytic system.  Attempts to change the chemical composition are not successful till now.

Minor corrections: 

i                  Please define all jargon for broad readers, for e.g., Page 2 Line 63: BET and line 65: BJH

Response: The full names of the methods used in determining the textural characteristics of the catalyst have been added to the text of the article (page 2, line 63: BET - Brunauer–Emmett–Teller and line 65: BJH - Barrett, Joyner and Halenda).

ii                 Page 2 line 66: typo “adsorbtion” 

Response: The typo was corrected.

iii               SI file Figure S1: Legend is missing in the figure for the two different graphs.

Response: The legend is added in Figure S1 (adsorption and desorption branches of the isotherm are labeled).

Reviewer 2 Report

In this work, Leonid Kustov et al. synthesized the catalyst Mo1-V0.3-Te0.13-Nb0.11-Ox by hydrothermal synthesis and characterized by a set of physicochemical methods (XRD, N2 adsorption, SEM, EDX). Direct microwave heating of the catalyst layer is proposed as an alternative way of energy-saving ethane-to-ethylene oxidation by a Mo-V-Te-Nb-Ox system. A substantial decrease of the reactor temperature upon the microwave-assisted process is accompanied by the extremely high catalyst selectivity, which remains at a very high level of 98+%. The contents are suitable for this journal, but major revisions are necessary.

1. The composition for the catalyst should be characterized by ICP-AES in comparison with the EDX result.

2. XPS should be conducted to characterize the chemical state of elements for the sample.

3. The stability of the catalyst is very important for the industrial application of the catalyst. The stability test should be added.

4. The relevant literatures should be cited, such as, Journal of Alloys and Compounds2022, 910,164745; Journal of Rare Earths 2022,40(5), 753-762; Chemical Engineering Journal 2022, 429,132388.

Author Response

In this work, Leonid Kustov et al. synthesized the catalyst Mo1-V0.3-Te0.13-Nb0.11-Ox by hydrothermal synthesis and characterized by a set of physicochemical methods (XRD, N2 adsorption, SEM, EDX). Direct microwave heating of the catalyst layer is proposed as an alternative way of energy-saving ethane-to-ethylene oxidation by a Mo-V-Te-Nb-Ox system. A substantial decrease of the reactor temperature upon the microwave-assisted process is accompanied by the extremely high catalyst selectivity, which remains at a very high level of 98+%. The contents are suitable for this journal, but major revisions are necessary.

Comment 1. The composition for the catalyst should be characterized by ICP-AES in comparison with the EDX result.

Response: We would like to thank the reviewer for the critical comments and useful advices and questions raised. We tried to do our best to improve our manuscript by taking into account all the comments. Below we give our responses to the comments. The changes in the text are highlighted in yellow. The results of determining the composition of the catalyst obtained by the ICP-AES method are in good agreement with the results obtained using the "surface" methods of EDX and XPS. The corresponding comment has been added to the text of the article (page 3).

Comment 2. XPS should be conducted to characterize the chemical state of elements for the sample.

Response: The results of the catalyst study by XPS method are added to the article (section 2.1., p. 3) and to the Supplementary Materials (Figure S4).  

Comment 3. The stability of the catalyst is very important for the industrial application of the catalyst. The stability test should be added.

Response: Long-term testing of the catalyst demonstrated that, after 10 hours of work, the conversion of ethane decreases from 30 to 28% without any selectivity loss.  

Comment 4. The relevant literatures should be cited, such as, Journal of Alloys and Compounds2022, 910,164745; Journal of Rare Earths 2022,40(5), 753-762; Chemical Engineering Journal 2022, 429,132388.

Response: Two articles indicated by the reviewer have been added to the list of references and cited in the introduction.

Le, T.M.N.; Checa, R.; Bargiela, P.; Aouine, M.; Millet J.M.M. New synthesis of pure orthorhombic Mo-V-A oxide phases, where A = Sb, Bi and Pb, and testing for the oxidation of light alkanes. Journal of Alloys and Compounds 2022, 910,164745. https://doi.org/10.1016/j.jallcom.2022.164745

Liu, B.; Yan, L.; Zhao, H.; Yang, J.; Zhao, J.; Song, H.; Chou, L. Role of cerium dopants in MoVNbO multi-metal oxide catalysts for selective oxidation of ethane. Journal of Rare Earths 2022, 40(5), 753-762. https://doi.org/10.1016/j.jre.2021.04.016

However, the third article indicated by the reviewer is a bit far from the subject of this article:

Chao Wan, Liu Zhou, Shuman Xu, Biyu Jin, Xin Ge Xing Qian Lixin Xu, Fengqiu Chen, Xiaoli Zhan, Yongrong Yang, Dang-duo Cheng, Defect engineered mesoporous graphitic carbon nitride modified with AgPd nanoparticles for enhanced photocatalytic hydrogen evolution from formic acid, Chemical Engineering Journal 2022, 429,132388. https://doi.org/10.1016/j.cej.2021.132388

Round 2

Reviewer 2 Report

This article can be accepted in the present form.